# Parental Feeding Styles and Risk of a New Carious Lesion in Preschool Children: A Longitudinal Study

**DOI:** 10.3390/nu15204387

**Published:** 2023-10-16

**Authors:** Jie Wen, Ye Tao, Liangyue Pang, Yina Cao, Huancai Lin, Yan Zhou

**Affiliations:** 1Hospital of Stomatology, Guanghua School of Stomatology, Sun Yat-sen University, Guangzhou 510055, China; wenj33@mail2.sysu.edu.cn (J.W.); taoye18@aliyun.com (Y.T.); pangly5@mail.sysu.edu.cn (L.P.); caoyn5@mail.sysu.edu.cn (Y.C.); linhc@mail.sysu.edu.cn (H.L.); 2Guangdong Provincial Key Laboratory of Stomatology, Sun Yat-sen University, Guangzhou 510055, China

**Keywords:** early childhood caries, new carious lesion, parental feeding styles, longitudinal study

## Abstract

Parents may influence children’s intake of sweets and sugar-sweetened beverages through their feeding styles. This study prospectively assessed the association between parental feeding styles and caries risk in preschool children. In November and December 2021, we recruited 1181 preschool children aged 3–4 years in Guangzhou, China, and then conducted a follow-up visit after 15 months. The children were examined for dental caries, and their parents answered the Chinese version of the Parental Feeding Style Questionnaire. Data were analyzed using logistic regression analysis. At baseline, ‘control over eating’ was found to be negatively associated with early childhood caries (OR = 0.60, 95%CI = 0.44, 0.83). After 15 months, no significant association between baseline parental feeding styles and the risk of a new carious lesion was found. When considering changes in parental feeding styles between the baseline and the follow-up visit, we found children with increasing scores on ‘instrumental feeding’ during the 15 months had a higher risk of a new carious lesion (OR = 1.91, 95%CI = 1.18, 3.12). However, children with always-intermediate scores (OR = 0.51, 95%CI = 0.30, 0.86), always-high scores (OR = 0.42, 95%CI = 0.23, 0.77), or increasing scores (OR = 0.50, 95%CI = 0.31, 0.83) on ‘control over eating’ had a lower risk of a new carious lesion. Our study highlighted the influence of parental feeding styles on caries risk in preschool children.

## 1. Introduction

Early childhood caries (ECC) refers to a child under 71 months with one or more decayed (noncavitated or cavitated lesions), missing (due to caries), or filled tooth surfaces in any primary tooth [1]. ECC is associated with acute and chronic pain, hospitalizations and emergency room visits, delays in growth and development, and diminished quality of life in young children [2]. At present, ECC is a global health problem, affecting almost half of preschool children. The prevalence of ECC is 48%, according to a meta-analysis of cross-sectional studies [3]. In China, the prevalence of ECC is more severe. According to the results of the Fourth National Oral Epidemiological Survey, the prevalence of ECC was 50.8%, 63.6%, and 71.9% for 3-, 4-, and 5-year-olds, respectively [4]. ECC is a complex multifactorial disease and is impacted by numerous factors, such as cariogenic microorganisms, frequent exposure to fermentable carbohydrates, poor oral hygiene habits, and socioeconomic status [2,5].

It has been well recognized that consumption of free sugars such as sugars added to food and beverage is of critical importance to the development of dental caries [6]. During the preschool period, parents play a crucial role in influencing the eating habits of their children. They may influence their children’s eating behavior through food parenting practices, which are food-specific behaviors or actions taken by parents to rear their children, affecting their children’s attitudes, behaviors, or beliefs [7]. 

Several instruments have been developed to measure food-related parenting practices, for example, the Child Feeding Questionnaire (CFQ) [8], the Preschooler Feeding Questionnaire (PFQ) [9], the Comprehensive Feeding Practices Questionnaire (CFPQ) [10], and the Parental Feeding Style Questionnaire (PFSQ) [11]. Among them, the PFSQ proved to have good reliability and validity, and there was a Chinese version [12]. The PFSQ was used to assess four aspects of parental feeding style, including ‘instrumental feeding’ (using food to modify a child’s behavior), ‘emotional feeding’ (using food to modify a child’s emotion), ‘prompting and encouragement to eat’ (encouraging children to consume a variety of foods) and ‘control over eating’ (controlling what, when, where and how much should the child consume) [11]. It involved three higher-order food parenting constructs—coercive control, structure, and autonomy support, proposed by Vaughn et al. [7]. ‘Instrumental feeding’ and ‘emotional feeding’ were classified as coercive control, which focuses on the parents’ goals without attention to the child’s satiety needs and may have an adverse impact on children’s eating, pushing and dominating the child to behave according to the parents’ desires. These two feeding styles were found to be positively correlated with the intake of high-energy-density food such as sweets and sugar-sweetened beverages (SSB) [13,14,15,16]. ‘Prompting or encouragement to eat’ was classified as autonomy support and ‘control over eating’ was classified as structure. As structure-related and autonomy-supporting parenting practices reflect parents’ approaches to support and guide their children to eat healthily while accounting for the children’s emotional and psychological needs, these food parenting practices may have a favorable impact on children’s eating [7]. Both feeding styles were associated with less consumption of high-energy-density food [13,14,15,16]. Therefore, parents may influence children’s intake of unhealthy food such as sweets and sugar-sweetened beverages through their feeding styles, which in turn are associated with the risk of caries. 

Based on the important role of diet in dental caries and the fact that young children’s diet is largely influenced by parental feeding styles, it is necessary to conduct a study using the PFSQ to explore the association between parental feeding styles and caries. Therefore, the objective of our study was to explore: (1) associations between parental feeding styles and the caries status of preschool children at baseline; (2) longitudinal associations between baseline parental feeding styles and the risk of a new carious lesion of preschool children after 15 months; (3) longitudinal associations between the change in parental feeding styles during the 15-month period and the risk of a new carious lesion after 15 months. The null hypothesis was that parental feeding styles would not be associated with the risk of a new carious lesion.

## 2. Materials and Methods

### 2.1. Study Design and Procedure

We recruited 3–4-year-old preschool children from seventeen kindergartens in Yuexiu District (urban area) and Zengcheng District (suburban area) of Guangzhou. Guangzhou is a mega-city in Guangdong Province in southern China, with a population of 18.8 million. The gross domestic product (GDP) was CNY 150,366 per capita in 2021 [17]. The water fluoride concentration was ≤0.3 mg/L [18]. The baseline data were collected in November and December 2021, including parental feeding styles, dental caries, oral hygiene status, body mass index (BMI), demographic characteristics, and oral health-related behaviors. Then we conducted a follow-up visit in February and March 2023 and collected information on dental caries and parental feeding styles. Throughout the follow-up interval, frequent lockdowns were occurring due to the pandemic of COVID-19. Children with severe systemic disease or who could not cooperate with dental examination were excluded. The study was approved by the Ethical Review Committee of Guanghua School of Stomatology, Sun Yat-Sen University (KQEC-2021-35-01). Written informed consent was obtained from each participant’s guardian before the study. 

### 2.2. Sample Size

According to Peduzzi, for logistic regression analysis, the number of events (death or illness) should be 5–10 times the number of independent variables [19]. At baseline, 19 independent variables were included in the regression model, and the caries prevalence in 3-year-old children was 50.8% according to the results of the Fourth National Oral Epidemiological Survey [4]; thus, the total sample size should be between 187 and 374. After 15 months, 20 independent variables were included in the regression model, and we postulated that at least 25% of children would have a new carious lesion; thus, considering a potential 20% loss to the follow-up rate, the total sample size should be between 500 and 1000. 

### 2.3. Clinical Examination

The dental examination was conducted in a bright classroom in kindergartens by three licensed dentists (Y.Z., Y.T., and LY.P.), using a portable light, community periodontal index (CPI) probe, and a disposable dental mirror. The oral hygiene status of the children was recorded using the visible plaque index (VPI) [20]. The presence or absence of visible plaque on the buccal surfaces (mesial/central/distal) and the lingual surfaces (central) of all teeth was recorded. The VPI score was calculated as the percentage of the number of surfaces with visible plaque relative to the total number of dental surfaces examined. Dental caries were measured using the decayed, missing, and filled teeth (dmft) index according to the World Health Organization caries diagnostic criteria [21]. The BMI of children was calculated as weight (in kilograms) divided by height (in meters) squared. Height and weight were measured by a qualified research assistant. The intra-examiner and inter-examiner reliability of caries diagnosis was evaluated by re-examining a 5% random sample of children on the same day. Over a 15-month period, an increase in dmft (Δdmft > 0) was defined as having a new carious lesion, and no increase in dmft (Δdmft = 0) was defined as having no new carious lesion.

### 2.4. Questionnaires

Parental feeding styles and caries-related factors were collected by a paper questionnaire, which was sent to the parents of children to fill out and collected by teachers. The questionnaire consisted of three parts, as follows: (1) Parental Feeding Style Questionnaire; (2) demographic characteristics: child gender, age (month), residence, family income, and parental education level; (3) oral health-related behaviors: frequency of eating deserts, drinking sugar-sweetened beverages and consumption of sweets before sleep; frequency of toothbrushing and supervised toothbrushing by parents; use of fluoride toothpaste and professional fluoride application within 6 months. In 2021, the per capita disposable income of urban residents in Guangzhou was CNY 74,416 (CNY ∼6200 per month) [22]. Households with a per capita monthly income of less than 6000 CNY were considered low-income, while households with a per capita monthly income of equal to or more than 10,000 CNY were considered high-income. Other households were considered moderate-income. 

The Chinese version of the PFSQ [12] was used to assess parental feeding styles, including ‘instrumental feeding’ (using food to modify a child’s behavior), ‘emotional feeding’ (using food to modify a child’s emotion), ‘prompting and encouragement to eat’ (encouraging children to consume a variety of foods), and ‘control over eating’ (controlling what, when, where and how much the child should consume). The PFSQ can be found in Appendix A. To allow Chinese mainland parents to better understand the items of the PFSQ, we converted the questionnaire into Simplified Chinese according to the opinion of epidemiologists. And then, we re-verified the reliability and validity of the PFSQ. Parents were asked to choose from a 5-point Likert scale. Response options included ‘never’ (=1), ‘rarely’ (=2), ‘sometimes’ (=3), ‘often’ (=4), ‘always’ (=5). A mean score was calculated for each subscale, where a higher score indicates a higher tendency for parents to adopt a particular style. 

To compare changes in parental feeding styles during the 15 months precisely, we divided the score on the subscale of the PFSQ into three categories. The top 27% and bottom 27% of scores were high scores and low scores, respectively. And, the rest were intermediate scores [23]. After 15 months, the same cut-off points as the baseline were used for classification. Thus, over a 15-month period, there were five variations in the parental feeding styles, that is, always low scores, always intermediate scores, always high scores, decreasing scores (i.e., from high and intermediate scores at baseline to intermediate and low scores after 15 months), increasing scores (i.e., from low and intermediate scores at baseline to intermediate and high scores after 15 months).

### 2.5. Statistical Analysis

Missing values in the PFSQ were computed by the participants’ mean of the subscale in case of ≤25% missing values per subscale. Otherwise, the participant was excluded from the analyses [24]. The five reversed items of ‘control over eating’ were re-coded before computing the average of the subscale. Internal reliability was assessed by Cronbach’s alpha for the four subscales. Exploratory factor analysis, using varimax rotation, was used to examine the factor structure of the PFSQ. Each parental feeding style score (i.e., the average score of the subscale) was the average of all items for that feeding style and was used for subsequent statistical description (e.g., mean and standard deviation) and statistical analysis. Chi-square tests (for unordered categorical variables), linear-by-linear association tests (for ordered categorical variables), and Student’s *t* tests (for continuous variables) were conducted to examine the baseline differences of the participants according to caries status at baseline (or the change in dmft after 15 months). A logistic regression model was used to examine (1) associations between parental feeding styles and caries status at baseline, adjusted for VPI, BMI, demographic characteristics, and oral health-related behaviors (model 1); (2) longitudinal associations between baseline parental feeding styles and risk of a new carious lesion after 15 months, adjusted for the covariates in model 1 plus past caries experience (model 2); (3) longitudinal associations between the change in parental feeding styles during the 15 months and risk of a new carious lesion after 15 months, adjusted for the covariates in model 2 (model 3). Individuals with missing data in covariates were excluded from the logistic regression analysis. All statistical analyses were performed using SPSS version 26.0 (IBM, Armonk, NY, USA). Statistical significance was set at *p* < 0.05 (two-sided).

## 3. Results

### 3.1. General Information

At baseline, a total of 1286 children were invited and, finally, 1181 eligible children were included in the study. Univariate analysis was performed for all 1181 children, and multivariate analysis was performed for 1158 children without missing values in covariates (model 1). After 15 months, a total of 1090 children (7.7% lost to follow-up) completed dental caries examination. Univariate analysis was performed for all 1090 children, and multivariate analysis was performed for 1074 children without missing values in covariates (model 2). Further, 1021 children (13.5% lost to follow-up) completed both dental caries examination and the PFSQ in the longitudinal study. Multivariate analysis was performed for all 1021 children (model 3) (Figure 1).

### 3.2. Internal Reliability of the Parental Feeding Style Questionnaire

Cronbach’s alpha coefficients for ‘instrumental feeding’, ‘emotional feeding’, ‘prompting or encouragement to eat’, ‘control over eating’, and all 27 items were 0.63, 0.84, 0.89, 0.67, 0.79, respectively. Ranges of the item-deleted Cronbach’s alphas are shown in Table 1. All Cronbach’s alpha coefficients of the subscale were greater than the item-deleted Cronbach’s alphas except item 6 of ‘control over eating’ (i.e., I allow my child to choose which foods to have for meals). If this item was deleted, the Cronbach’s alpha coefficient increased from 0.67 to 0.72. Nevertheless, the initial Cronbach’s alpha coefficient of this subscale was 0.67, still higher than the 0.63 reported in the previous study [12], so we retained item 6.

### 3.3. Construct Validity of the Parental Feeding Style Questionnaire

Factor analysis revealed six factors for the PFSQ that accounted for 59.6% of variance (Table 2). A one-factor structure was revealed for ‘instrumental feeding’, ‘emotional feeding’, and ‘promoting and encouragement to eat’, respectively, which was consistent with the original factor structure of the PFSQ [11]. However, a three-factor structure was revealed for ten items of ‘control over eating’. The result was similar to those of Tam et al. [12], who initially found a three-factor structure of ‘control over eating’ and finally confirmed it as a two-factor structure. Considering that the factor structure in the current sample was similar to the original PFSQ factor structure and the PFSQ had been verified in China, we performed further statistical analysis of the four subscales defined by Wardle et al. [11], in order to allow comparison with previous research utilizing the PFSQ.

### 3.4. Reliability of Caries Diagnosis

For the caries diagnosis, the inter-examiner kappa values ranged from 0.84 to 0.88 and the intra-examiner kappa values ranged from 0.88 to 0.95 at baseline. At follow-up after 15 months, the inter-examiner kappa values ranged from 0.81 to 0.84 and the intra-examiner kappa values ranged from 0.82 to 0.92.

### 3.5. Association between Parental Feeding Styles and Caries Status at Baseline

A total of 1181 children participated in the study at baseline. Of them, 52.1% were boys and 47.9% were girls. Of the participants, 54.0% lived in the urban area and 46.0% lived in the suburban area. The mean age of the participants was 44.39 ± 3.58 months. More than three-quarters of the participating parents were highly educated (i.e., college or above). The prevalence of caries was 54.4% and the mean dmft was 2.80. Most of the PFSQs were completed by the mother (79.9%), and others were filled out by the father (20.1%). The total score for each subscale was 5 and the mean scores for ‘instrumental feeding’, ‘emotional feeding’, ‘prompting or encouragement to eat’, and ‘control over eating’ were 2.59, 2.27, 3.90, and 3.67, respectively.

Characteristics of the participants according to caries status at baseline are presented in Appendix A. All four parental feeding styles and all variables except BMI, gender, using fluoride toothpaste, and professional fluoride application within 6 months differed significantly between children with and without caries. 

In a multivariate logistic regression model, we found an association between parental feeding styles and caries (Table 3). After adjusting for covariates, only ‘control over eating’ was negatively associated with caries (OR = 0.60, 95%CI = 0.44, 0.83), while other parental feeding styles were not significantly related to caries.

VPI, older age, living in a suburban area, low educational level of the child’s mother, drinking SSB once a week, and frequent consumption of sweets before sleep were positively associated with caries, whereas no professional fluoride application within 6 months was negatively associated with caries.

### 3.6. Longitudinal Associations between Baseline Parental Feeding Styles and Risk of a New Carious Lesion after 15 Months 

After 15 months, 70.7% of children had dental caries, and the mean dmft was 4.63. During the 15-month follow-up period, 58.8% of children had an increase in dmft (i.e., Δdmft > 0). Among them, 35.6% of children without past caries experience had a new carious lesion, while 78.2% of children with past caries experience had a new carious lesion. Characteristics of the participants according to the change in dmft after 15 months are presented in Appendix A. ‘Prompting or encouragement to eat’ and ‘control over eating’ at baseline differed significantly among those with and without an increase in dmft, while ‘instrumental feeding’ and ‘emotional feeding’ showed no significant difference. Other variables except BMI, gender, and using fluoride toothpaste differed significantly among those with and without an increase in dmft.

In a multivariate logistic regression model, no association between parental feeding styles and risk of a new carious lesion was found after adjustment (Table 4). The strongest predictor of a new carious lesion was past caries experience (OR = 5.58, 95%CI = 4.16, 7.48). Older age, higher VPI scores, and occasionally or never-supervised toothbrushing by parents were positively associated with a risk of a new carious lesion.

### 3.7. Longitudinal Association between the Change in Parental Feeding Styles during the 15 Months and Risk of a New Carious Lesion after 15 Months

We explored the association between the change in parental feeding styles during the 15 months and the risk of a new carious lesion after 15 months (Table 5). After adjusting for confounding factors, we found that children with increasing scores on ‘instrumental feeding’ during the 15 months had higher risk of a new carious lesion than those with always-low scores (OR = 1.91, 95%CI = 1.18, 3.12). On the contrary, children with always-intermediate scores (OR = 0.51, 95%CI = 0.30, 0.86), always-high scores (OR = 0.42, 95%CI = 0.23, 0.77), or increasing scores (OR = 0.50, 95%CI = 0.31, 0.83) on ‘control over eating’ had a lower risk of a new carious lesion than those with always-low scores.

## 4. Discussion

This study was a longitudinal study to investigate the relationship between parental feeding styles and ECC. We used the PFSQ to assess four aspects of feeding style and explore their association with ECC. The PFSQ was first developed by Wardle et al. [11], then it had been translated into Chinese and proved to have good reliability and validity [12]. Our results showed that the reliability and validity of the PFSQ were similar to those reported in the previous study in Hong Kong [12]. This suggested that the PFSQ was suitable for use in Guangzhou, which was geographically close to Hong Kong and located in southern China. 

Parental feeding styles refer to specific practices that influence the food intake of children. Previous studies have demonstrated an association between parental feeding styles and children’s intake of high-energy-density food, caries, and body weight [13,14,15,16,25,26]. In our study, the mean scores for ‘instrumental feeding’, ‘emotional feeding’, ‘prompting or encouragement to eat’, and ‘control over eating’, were 2.59, 2.27, 3.90, and 3.67, respectively. This indicated that parents more often used ‘prompting or encouragement to eat’ and ‘control over eating’ than ‘instrumental feeding’ and ‘emotional feeding’. 

At baseline, we found an association between ‘control over eating’ and ECC. After adjusting for confounding factors, children whose parents had a higher tendency to use ‘control over eating’ had lower risk of caries. ‘Control over eating’ was understood to occur when parents set clear expectations and boundaries regarding what, when, where, and how much their children should eat [7,27]. These rules and limits set by parents could reduce the intake of high-energy-density food such as sweets and SSB in children [13,14,15]. With such a reduction, microflora do not transform enough sugar to acid, and this ultimately reduces the risk of caries [28]. However, the result of a previous study was different, which reported that ‘control over eating’ contributed to an increase in dental caries [25]. Cultural differences, different study designs, and failure to adjust for confounding factors might account for these differences. Poor oral hygiene was the risk indicator associated with the presence of cavitated lesions in preschool children, which could be reflected by the plaque presence on tooth surfaces [29,30]. Our study also found an association between VPI and caries cross-sectionally and longitudinally. In addition, older age, living in a suburban area, low educational level of the child’s mother, drinking SSB once a week, and frequent consumption of sweets before sleep were found to be associated with the presence of ECC. This was similar to the results of a recent provincial cross-sectional survey conducted in Guangdong Province [31]. Interestingly, we found that children who had professional fluoride application within 6 months had a higher prevalence of ECC than those without professional fluoride application. This difference might be due to the different patterns of dental visits in China, where parents took their children to visit dentists only when the children had already experienced oral health problems [31].

No association was found between parental feeding styles at baseline and risk of a new carious lesion after 15 months, maybe because past caries experience significantly attenuated this link. Past caries experience was associated with a cariogenic oral microbiome composition, a cariogenic diet, and other factors that increased the risk of developing a new carious lesion [32]. Thus, it was not surprising that individuals with past caries experience had an odds ratio of 5.58 for developing a new carious lesion, overwhelming the associations observed with parental feeding styles. 

Previous studies have found an association between the trajectory of sugar consumption and dental caries at 48 months in children. Children with increasing sugar consumption or always-high sugar consumption had a higher prevalence of caries [33]. Given that parental feeding styles might change over time, their association with caries might be dynamic. Our results showed that parents increasing their tendency to adopt ‘instrumental feeding’ during the 15 months was associated with a higher risk of a new carious lesion in their children. ‘Instrumental feeding’ reflected parents’ use of food to control children’s behavior. In this case, sweet and energy-rich foods were the first choice because they were in line with children’s genetically predetermined predispositions [34]. Previous studies supported this hypothesis, reporting a positive association between ‘instrumental feeding’ and the high-energy-density food intake of children [13,14,15,16]. Interestingly, ‘instrumental feeding’ has also been found to be associated with higher weight over time [27]. Parental feeding styles influenced the dietary intake of young children and were associated with both caries and weight gain. More attention should be paid to parental feeding styles. 

However, it was unclear why both ‘instrumental feeding’ and ‘emotional feeding’ were associated with increased unhealthy food intake, but we only found an association between ‘instrumental feeding’ and caries outcomes over time. It has been hypothesized that ‘emotional feeding’ can only be used when a child is upset or emotional and therefore might occur less frequently [27]. The potential higher frequency of ‘instrumental feeding’ could explain why ‘instrumental feeding’, but not ‘emotional feeding’, was associated with the risk of a new carious lesion in children over time.

We also found that parents having a higher tendency or an increasing tendency to adopt ‘control over eating’ during the 15 months was associated with a lower risk of a new carious lesion in their children. This was consistent with our previous cross-sectional results. Furthermore, ‘control over eating’ was likely to have a long-term effect on children. Newly adopted food rules might be less effective for children because they are not accustomed to rules and limits in general. However, with consistent use over time, children can learn the rules and develop good eating habits [7]. Although ‘prompting and encouragement to eat’ was also associated with a healthy diet in children, our study did not find its association with caries outcomes. Nembhwani et al. did not find an association between ‘prompting and encouragement to eat’ and ECC as well, supporting our results [25].

Unexpectedly, we found that 58.8% of children had a new carious lesion after 15 months. Even among children without past caries experience, 35.6% had a new carious lesion, which exceeded our initial estimate of 25%. Due to the pandemic of COVID-19, children had often been locked down at home for the previous 15 months. They might change their eating, drinking, and toothbrushing habits during lockdown, and, thus, increase their risk of developing caries [35,36]. In addition, frequent lockdowns due to the pandemic of COVID-19 caused the children to stay at home with their parents for a prolonged time. Therefore, parental feeding styles were likely to have a greater impact on their children. A recent study reported that during the COVID-19 pandemic, parents experienced increased levels of stress. Higher stress was associated with more nonnutritive use of food and snacks (e.g., ‘emotional feeding’ and ‘instrumental feeding’) [37]. According to the results of our study, children with increasing scores on ‘instrumental feeding’ had a higher risk of a new carious lesion. This might be one of the reasons for the high risk of a new caries lesion in our study population.

Our findings provide new insights into the intervention of caries in young children through the feeding style of their parents. Parents should set appropriate rules and limits to promote healthy food intake for children. In addition, parents should avoid using food to control children’s behavior.

This study had several strengths. First, the study had a large sample size and balanced gender and urban/suburban areas. Second, our study was a prospective cohort study. It could better explore the association between the change in parental feeding styles and the risk of a new carious lesion. However, there were still limitations in this study. Our study was conducted in a mega-city, so the results for this district were applicable only to other regions with similar demographics. In addition, longer follow-ups would allow us to draw more robust conclusions.

## 5. Conclusions

Our study highlighted the importance of parental feeding styles and revealed their associations with the risk of a new carious lesion in preschool children. An increase in parents’ tendency to adopt ‘instrumental feeding’ during the 15 months was associated with a higher risk of a new carious lesion in their children. However, parents having a higher tendency or an increasing tendency to adopt ‘control over eating’ during the 15 months was associated with a lower risk of a new carious lesion in their children. 

## Figures and Tables

**Figure 1 nutrients-15-04387-f001:**
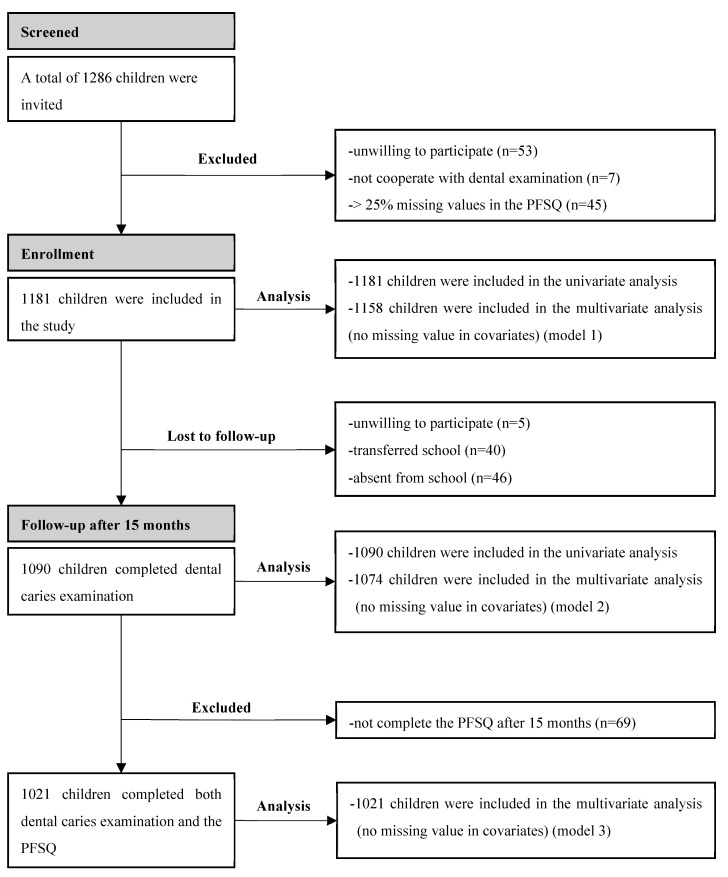
Flowchart of the study.

**Table 1 nutrients-15-04387-t001:** Internal reliability of the Parental Feeding Style Questionnaire.

Parental Feeding Style Questionnaire	Mean Score (SD)	Cronbach’s Alpha	Item-Deleted Cronbach’s Alpha
Instrumental feeding (4 items)	2.59 (0.59)	0.63	0.51–0.62
Emotional feeding (5 items)	2.27 (0.67)	0.84	0.79–0.83
Prompting or encouragement to eat (8 items)	3.90 (0.68)	0.89	0.86–0.89
Control over eating (10 items)	3.67 (0.48)	0.67	0.62–0.72
Total (27 items)		0.79	0.77–0.80

SD: standard deviation.

**Table 2 nutrients-15-04387-t002:** Exploratory factor analysis using the Parental Feeding Style Questionnaire.

	Factor 1	Factor 2	Factor 3	Factor 4	Factor 5	Factor 6
Instrumental feeding						
In order to get my child to behave him/herself I promise him/her something to eat.	−0.06	0.21	0.03	−0.15	**0.72**	−0.02
If my child misbehaves I withhold his/her favourite food.	0.04	0.13	0.11	0.07	**0.63**	0.01
I use puddings as a bribe to get my child to eat his/her main course.	−0.11	0.27	−0.06	−0.36	**0.45**	−0.19
I reward my child with something to eat when s/he is well behaved.	0.12	0.24	−0.01	0.01	**0.74**	0.07
Emotional feeding						
I give my child something to eat to make him/her feel better when s/he is feeling upset.	0.06	**0.63**	0.04	−0.03	0.40	0.00
I give my child something to eat to make him/her feel better when s/he has been hurt.	0.02	**0.78**	0.03	−0.04	0.23	−0.02
I give my child something to eat if s/he is feeling bored.	−0.06	**0.75**	0.02	−0.18	0.09	−0.03
I give my child something to eat to make him/her feel better when s/he is worried.	0.03	**0.83**	0.02	−0.12	0.14	−0.02
I give my child something to eat to make him/her feel better when s/he is feeling angry.	0.00	**0.74**	−0.12	−0.19	0.11	−0.07
Promoting and encouragement to eat						
I encourage my child to look forward to the meal.	**0.68**	0.08	0.13	0.09	−0.01	0.12
I praise my child if s/he eats what I give him/her.	**0.65**	0.06	0.09	−0.04	0.18	−0.17
I encourage my child to eat a wide variety of foods.	**0.75**	−0.04	0.15	−0.02	0.02	0.15
I present food in an attractive way to my child.	**0.64**	0.14	0.15	−0.04	−0.12	−0.05
I encourage my child to taste each of the foods I serve at mealtimes.	**0.78**	−0.09	0.20	0.06	−0.01	0.19
I encourage my child to try foods that s/he hasn’t tasted before.	**0.75**	−0.11	0.18	0.00	−0.01	0.21
I encourage my child to enjoy his/her food.	**0.78**	−0.02	0.15	0.09	−0.06	0.15
I praise my child if s/he eats a new food.	**0.78**	−0.02	0.18	0.01	0.12	−0.01
Control over eating						
I decide when it is time for my child to have a snack.	0.24	0.05	**0.78**	−0.04	0.08	−0.01
I decide how many snacks my child should have.	0.24	−0.08	**0.78**	0.04	0.12	0.08
I decide what my child eats between meals.	0.27	0.03	**0.77**	0.02	−0.03	0.03
I decide the times when my child eats his/her meals.	0.29	−0.04	**0.62**	0.08	−0.02	0.27
I insist my child eats meals at the table.	0.32	−0.07	0.24	0.26	0.01	**0.67**
I allow my child to choose which foods to have for meals *.	−0.12	0.03	−0.11	**0.35**	−0.03	−0.65
I allow my child to wander around during a meal *.	0.11	−0.13	−0.04	**0.60**	−0.08	0.47
I allow my child to decide when s/he has had enough snacks to eat *.	−0.03	−0.14	−0.04	**0.66**	0.00	−0.20
I let my child eat between meals whenever s/he wants *.	0.08	−0.07	0.10	**0.76**	−0.06	0.00
I let my child decide when s/he would like to have her meal *.	−0.03	−0.16	0.02	**0.71**	−0.01	0.04

* Reversed questions. Bold numbers indicate the factor with the highest load.

**Table 3 nutrients-15-04387-t003:** Multivariate logistic regression results of associations between parental feeding styles and caries status at baseline (*n* = 1158).

	OR (95%CI)	*p*
Parental feeding style		
Instrumental feeding	1.13 (0.88–1.46)	0.329
Emotional feeding	0.97 (0.78–1.22)	0.806
Prompting or encouragement to eat	0.87 (0.70–1.07)	0.174
Control over eating	0.60 (0.44–0.83)	**0.002**
Age (months)	1.08 (1.04–1.12)	**<0.001**
BMI	1.01 (0.90–1.12)	0.924
Visible plaque index ^a^	1.29 (1.20–1.39)	**<0.001**
Gender		
Female (Ref: male)	1.02 (0.79–1.33)	0.854
Residence		
Suburban (Ref: urban)	1.61 (1.20–2.15)	**0.001**
Paternal education level		
High school or below (Ref: college or above)	1.25 (0.84–1.86)	0.280
Maternal education level		
High school or below (Ref: college or above)	1.79 (1.19–2.69)	**0.005**
Household monthly income		
High-income (Ref: low-income)	0.81 (0.59–1.11)	0.194
Moderate-income (Ref: low-income)	0.89 (0.64–1.24)	0.498
Frequency of eating deserts		
≥2 times per week (Ref: <1 time per week)	1.29 (0.92–1.81)	0.143
1 time per week (Ref: <1 time per week)	1.24 (0.82–1.88)	0.300
Frequency of drinking sugar-sweetened beverages		
≥2 times per week (Ref: <1 time per week)	1.43 (0.85–2.41)	0.180
1 time per week (Ref: <1 time per week)	1.50 (1.00–2.23)	**0.049**
Consumption of sweets before sleep		
Frequently (Ref: never)	1.47 (1.00–2.15)	**0.048**
Occasionally (Ref: never)	1.22 (0.91–1.64)	0.190
Frequency of toothbrushing		
<2 times per day (Ref: ≥2 times per day)	0.85 (0.65–1.12)	0.258
Supervised toothbrushing		
Occasionally or never (Ref: frequently)	0.87 (0.67–1.14)	0.323
Using fluoride toothpaste		
No (Ref: yes)	0.94 (0.69–1.28)	0.680
Unknown (Ref: yes)	0.87 (0.61–1.23)	0.425
Professional fluoride application within 6 months		
No (Ref: yes)	0.73 (0.53–0.98)	**0.038**
Unknown (Ref: yes)	0.84 (0.50–1.40)	0.499

OR: odds ratio, CI: confidence interval. ^a^ Visible plaque index was calculated as a percentage of the number of surfaces with plaque to the total number of surfaces examined, and the odds ratio of the visible plaque index value increasing by 0.1 is presented. *p*-value in bold indicates statistical significance.

**Table 4 nutrients-15-04387-t004:** Multivariate logistic regression results of association between baseline parental feeding styles and risk of a new carious lesion after 15 months (*n* = 1074).

	OR (95%CI)	*p*
Parental feeding style		
Instrumental feeding	1.05 (0.79–1.39)	0.749
Emotional feeding	0.94 (0.73–1.20)	0.609
Prompting or encouragement to eat	0.93 (0.73–1.17)	0.537
Control over eating	0.88 (0.62–1.25)	0.485
Age (months)	1.04 (1.00–1.09)	**0.044**
BMI	0.99 (0.88–1.11)	0.867
Visible plaque index ^a^	1.11 (1.03–1.21)	**0.010**
Past caries experience		
Caries (Ref: caries-free)	5.58 (4.16–7.48)	**<0.001**
Gender		
Female (Ref: male)	1.00 (0.75–1.32)	0.991
Residence		
Suburban (Ref: urban)	1.13 (0.81–1.57)	0.470
Paternal education level		
High school or below (Ref: college or above)	1.32 (0.83–2.08)	0.236
Maternal education level		
High school or below (Ref: college or above)	0.79 (0.50–1.25)	0.311
Household monthly income		
High-income (Ref: low-income)	0.93 (0.66–1.31)	0.683
Moderate-income (Ref: low-income)	1.13 (0.78–1.62)	0.519
Frequency of eating deserts		
≥2 times per week (Ref: <1 time per week)	1.21 (0.83–1.77)	0.322
1 time per week (Ref: <1 time per week)	0.70 (0.45–1.10)	0.118
Frequency of drinking sugar-sweetened beverages		
≥2 times per week (Ref: <1 time per week)	0.90 (0.50–1.59)	0.707
1 time per week (Ref: <1 time per week)	1.05 (0.67–1.65)	0.821
Consumption of sweets before sleep		
Frequently (Ref: never)	0.94 (0.62–1.42)	0.771
Occasionally (Ref: never)	1.02 (0.74–1.42)	0.896
Frequency of toothbrushing		
<2 times per day (Ref: ≥2 times per day)	1.21 (0.89–1.64)	0.218
Supervised toothbrushing		
Occasionally or never (Ref: frequently)	1.45 (1.08–1.95)	**0.015**
Using fluoride toothpaste		
No (Ref: yes)	0.92 (0.65–1.29)	0.621
Unknown (Ref: yes)	0.77 (0.53–1.12)	0.174
Professional fluoride application within 6 months		
No (Ref: yes)	0.88 (0.63–1.23)	0.463
Unknown (Ref: yes)	1.39 (0.78–2.47)	0.268

OR: odds ratio, CI: confidence interval. ^a^ Visible plaque index was calculated as a percentage of the number of surfaces with plaque to the total number of surfaces examined, and the odds ratio of the visible plaque index value increasing by 0.1 is presented. *p*-value in bold indicate statistical significance.

**Table 5 nutrients-15-04387-t005:** Multivariate logistic regression results of association between the change in parental feeding styles during the 15 months and risk of a new carious lesion after the 15 months (*n* = 1021).

	Instrumental Feeding (OR, 95%CI)	Emotional Feeding (OR, 95%CI)	Prompting or Encouragement to Eat (OR, 95%CI)	Control over Eating (OR, 95%CI)
Increasing scores	**1.91 (1.18–3.12) ****	1.46 (0.87–2.44)	0.87 (0.52–1.48)	**0.50 (0.31–0.83) ****
Decreasing scores	1.17 (0.75–1.83)	1.13 (0.68–1.89)	0.68 (0.41–1.14)	0.68 (0.41–1.14)
Always-high scores	1.18 (0.67–2.09)	0.96 (0.54–1.68)	1.49 (0.82–2.73)	**0.42 (0.23–0.77) ****
Always-intermediate scores	1.29 (0.77–2.17)	1.33 (0.76–2.31)	0.85 (0.50–1.46)	**0.51 (0.30–0.86) ***
Always-low scores	1.00	1.00	1.00	1.00

OR: odds ratio, CI: confidence interval. *: *p* < 0.05; **: *p* < 0.01. Numbers in bold indicate statistical significance.

## Data Availability

All data generated or analyzed during this study were included in this article. Further inquiries can be directed to the corresponding author.

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
