# Peer review of "Parental Feeding Styles and Risk of a New Carious Lesion in Preschool Children: A Longitudinal Study"

_nutrients, 2023, doi:10.3390/nu15204387_

Round 1
Reviewer 1 Report
[General]
The aim of this manuscript is to clarify the relation between the parental feeding style and risk of children caries by a longitudinal study. Parental feeding styles were examined using PFSQ. The validity of the PFSQ was analyzed using exploratory factor analysis. And, the authors analyzed the influence of changes in parental behavior over a 15-month period on the risk of dental caries by behavioral style. The authors showed that in the group with increased PFSQ scores, "instrumental feeding" was a factor that increased caries, while "control over eating" was a factor that suppressed the occurrence of caries. Thus, the results presented by the authors are clear. However, the method of analysis (especially the handling of score values) is unclear. Therefore, several descriptions must be improved.
[Major points]
1. About questionnaire: The contents of questionnaires (PSFQ) are shown in Table 2. It is better to first present the questionnaire (and answers if possible) as a separate table. I think this will make it easier for readers to understand the contents.
2. About questionnaire: The authors used 5-point Likert scale. I think this method is a common method, but it is better to list all the options instead of just "never" and "always". Moreover, considering that the average score values are calculated in Table 1, it is necessary to clearly state how these responses were quantified when performing analysis(ex. never = 1 and always = 5).
3. Table 1: The auhors calculated SD of score for each parental each style.
Please specify how you calculated this SD. Is it SD of the data for 4 items (in instrumental feeding) multiplied by the number of people?
4. Table 2: All values should be displayed. It is better to display large numbers in bold.
5. Logistic regression: The method should clearly state how the PSFQ values were used in the analysis (sum or average). Although this does not change the P values, it changes the values of the odds ratio and confidence interval, which may change the interpretation of the results (Tables 3 to 5).
6. Table 3 (and also Table 4): The odds ratio in logistic regression analysis indicates how many times the odds of the response variable increase when the value of the explanatory variable increases by 1. The authors showed that the odds ratio of visible plaque index (VPI) is 12.70 (95% CI: 5.95-27.09) in Table 3. The means of VPI are 0.34 (no increase in dmft) and 0.42 (increase in dmft), and the difference is 0.08. Therefore, it is thought that the odds ratio is obtained as a larger value than the actual value.
Please consider presenting the odds ratio when the VPI value increases 0.1. If the VPI value increases by 0.1, The odds ratio is 1.29 (95% CI: 1.20-1.39) (12.70^0.1 = 1.29; 5.95^0.1 = 1.20; 27.09^0.1 = 1.39).
7. (Page 13, line 311): The authors described "individuals with ... have a more than 5-fold increased risk ..." from the results that odds ratio of past caries experience is 5.58 in Table 4. This is a common misinterpretation. When the prevalence of the disease is small, the odds ratio is close to the value of the risk ratio. In this case, the authors' interpretation would be correct. However, because the prevalence of caries is high in this study, such interpretation is inappropriate. It should be written as "odds ratio is 5.58".
[Minor points]
1. (Not required) Tables S1 and S2: The authors analyzed chi-square test for the comparison between categorical variables. For ordered categorical variables (household monthly income, frequency of each deserts, frequency of drinking sugar-sweetened beverages, and consumption of sweets before sleep), linear-by-linear association tests are more appropriate although it is not wrong to use the chi-square test. I recommend the usage of linear-by-linear association test instead of chi-square test.
2. typo: (5) is (3) (Page 4, line 161).
Reviewer 2 Report
Parental feeding styles and risk of a new carious lesion in pre- school children: a longitudinal study
This study prospectively assessed the association between four of parental feeding styles (‘instrumental feeding’, ‘emotional feeding’, ‘prompting and encouragement to eat’ and ‘control over eating’) and caries risk in preschool children aged 3-4 years in Guangzhou, China, and then conducted a follow-up visit after 15 months. After 15 months, no significant association between baseline parental feeding styles and risk of a new carious lesion was found.
General remarks:
English must be ameliorated. Examples: in the abstract: “we found children whose parents with increasing scores on ‘instrumental feeding’ during 15 months…”, line 21:“While children whose parents with always intermediate scores…”, line 314 : “. Children with increasing and always high sugar consumption have the highest prevalence of caries”.
Introduction
This section is written well, with relevant references to prior studies.
A few points to correct:
“Early childhood caries (ECC) refers to a child under 71 months with one or more decayed (with or without cavitation lesions)…” -decayed is with cavitation, not both with or without.
When you say “In China, the prevalence of ECC is more severe”, you need to give the numbers (percentage) in other countries so the reader can compare.
Household monthly income – the numbers are not relevant- please switch to “high”, “moderate” etc.
Methods
This section is written well.
Supplementary Table S1: Needs to be divided into two separate tables: one for demographic data and the other for Parental feeding style, diet and dental behavoiur.
Household monthly income – the numbers are not relevant- please switch to “high”, “moderate” etc.
Discussion
This section is written well, with relevant references to prior studies. The last paragraph explains that the study was actually conducted through the COVID-19, with the children being often been locked down at home. This is a very interesting point which and may have influenced the study. I think the authors need to add this fact in the Methods section, and elaborate on it some more in the Discussion.
English must be ameliorated. Examples: in the abstract: “we found children whose parents with increasing scores on ‘instrumental feeding’ during 15 months…”, line 21: “While children whose parents with always intermediate scores…”, line 314 : “. Children with increasing and always high sugar consumption have the highest prevalence of caries”.
Reviewer 3 Report
I appreciate the opportunity to review this prospective study, which aimed to assess the association between parental feeding styles and caries risk in preschool children. The study is intriguing, and while the topic has been previously researched, it benefits from a substantial sample size. The study is well-executed and effectively written, but there are a few necessary improvements:
Introduction: The introduction is well-crafted and effectively introduces the research issue. To enhance clarity, it is recommended that the authors include the null hypothesis of the study at the end of the introduction.
Fig1. Flowchart: Visually revise Figure 1 to ensure clarity and readability.
Strengths and Limitations: The authors should explicitly outline the strengths and limitations of the study. Strengths could include aspects like the large sample size, robust methodology, or novel findings. In contrast, limitations should address potential weaknesses or constraints, such as sampling biases or limitations in data collection methods. This information will provide a more comprehensive perspective on the study's validity and generalizability.
By addressing these points, the study can further enhance its contribution to the field of caries risk assessment in preschool children and provide a clearer understanding of its significance.
Round 2
Reviewer 1 Report
The authors appropriately corrected the manuscript in response to my all comments.